# Particulate Matter Induced Adverse Effects on Eye Development in Zebrafish (*Danio rerio*) Embryos

**DOI:** 10.3390/toxics12010059

**Published:** 2024-01-11

**Authors:** Dalawalla G. Charith E. Priyadarshana, Jayeon Cheon, Yoonsung Lee, Seon-Heui Cha

**Affiliations:** 1Department of Integrated Bioindustry, Hanseo University, Seosan-si 31962, Republic of Korea; 2Department of Marine Bio and Medical Sciences, Hanseo University, Seosan-si 31962, Republic of Korea; wkdus4360@naver.com; 3Clinical Research Institute, Kyung Hee University Hospital at Gangdong, College of Medicine, Kyung Hee University, Seoul 05278, Republic of Korea

**Keywords:** particulate matter, toxicity, zebrafish, eye disease model

## Abstract

Particulate matter (PM) can cause human diseases, particularly respiratory diseases. Since eyes are directly exposed to the air, they might be directly adversely affected by PM. Therefore, we determined the toxicity caused to eye development by PM using zebrafish (*Danio rerio*) embryos. The PM-induced embryo toxicity was dependent on dose and time and caused significant morphological defects, reducing the total body length and the total eye area. Reactive oxygen species (ROS) overproduction was confirmed in the PM treatment group, and antioxidant genes (*cat* and *sod2*), photoreceptor cell development, pigmentation genes (*atoh8*, *vsx1*, and *rho*), eye-embryogenesis genes (*pax6a* and *pax6b*), and eye-lens-development genes (*cryaa*) were downregulated, while eye-development genes (*crybb1*) were upregulated. In conclusion, PM had a direct adverse effect on the eyes, and zebrafish embryos can be used as a model to evaluate PM-induced eye toxicity *in vivo*.

## 1. Introduction

Air pollution is becoming a severe environmental hazard to human health [1]. The composition of air pollution varies greatly depending on the source, but clinical studies indicate that particulate matter (PM) has a worse effect on health than the gaseous components of air pollution [2]. The World Health Organization (WHO) has noted that air pollution caused by PM contributes to more than 3.5 million premature deaths per year [3]. According to the aerodynamic diameter, PM is categorized into coarse particles (between 2.5 and 10 μm; PM_10_) and fine particles (less than 2.5 μm; PM_2.5_), which comprise millions of solid particles (metals), various gas particles, organic and inorganic hazardous materials (polyaromatic hydrocarbons, PAHs), and liquid droplets that collectively pose risks to human health [4,5]. And it has been reported that PM_2.5_ is much more harmful than PM_10_ to the human body [6]. Epidemiological studies have illustrated that cardiovascular diseases, respiratory tract diseases (lung cancer), cerebrovascular diseases, and skin diseases can occur due to PM exposure [7,8,9]. Moreover, it has been reported that PM-containing fine particles pose severe health risks due to their ability to infiltrate blood circulation while damaging deep sensitive tissues in the lungs and brain [10]. PM induces oxidative stress and inflammation in lung tissues, leading to severe adverse effects that cause diseases like lung cancer [11,12]. Although it has been reported that PM has a serious impact on human health, the composition of PM is diverse, and it has not yet been fully established how harmful it is or at what concentrations it presents the highest risk.

Unlike internal organs, the eyes, in particular, are directly exposed to air pollutants, which may present a greater risk of adverse effects on the eye, leading to the development and progression of diseases [13]. Acute and chronic exposure to air pollution has been linked to eye conditions such as itchiness, irritation, and excessive tearing [14,15,16]. Some studies have reported that allergic conjunctivitis is significantly related to air pollution, as the number of reported cases increased with exposure to PM dose-dependently. Most of these are based on epidemiological studies or other indirect evidence suggesting that the presence of PM caused diseases [17,18]. Moreover, there are not enough studies that evaluate the effects of PM during eye development and disease progression. Therefore, it is vital to assess the toxic effects of PM on the eye to prevent and manage human eye diseases caused by environmental pollution.

The zebrafish (*Danio rerio*) is a widely used model organism for assessing cardiovascular, neurological, intestinal, and hepatic-related toxicity, and studies use zebrafish embryos due to their advantages, which include transparency, low-cost husbandry, and genetic comparability with the human genome [9,19,20]. In addition, the zebrafish eye is similar to the human eye in terms of morphology, physiology, gene expression, and function; thus, most of the recent literature has evaluated the adverse effects of PM using the zebrafish model and has primarily focused on cardiovascular and respiratory toxicities and disease development [21]. Based on the previous research results and insights provided via the study of major eye diseases through reverse genetic techniques [22], it is expected that zebrafish could be used as an eye disease model system. In this study, therefore, we evaluated the eye development and toxicity in zebrafish embryos after PM exposure to establish an alternative model system for evaluating environmental pollutants such as PM.

## 2. Materials and Methods

### 2.1. Chemicals and Reagents

Particulate matter (PM, certified reference material no. 28) was purchased from the National Institute for Environmental Studies (NIES, Ibaraki, Japan). 2′,7′-dichlorodihydrofluorescein diacetate (H_2_DCF-DA) was purchased from Thermo Fisher Scientific (Waltham, MA, USA). All other chemical reagents used in this study were obtained from commercial sources, with more than 99.9% purity levels. 

### 2.2. Zebrafish Husbandry and Embryo Collection

Wild-type AB strain adult zebrafish were obtained from the Fluorescent Reporter Zebrafish Cooperation Center (FRZCC 1072) of Korea University, and zebrafish husbandry was performed according to a previously described method [23]. Adult male and female zebrafish were raised under standard 14/10, light/dark cycle at 28.5 °C in a water-circulated system and were fed twice a day. Healthy male and female zebrafish were placed in the mating tank at a ratio of 2:1 and allowed to lay eggs. On the morning of the next day, fertilized eggs were collected into a 90 mm diameter petri dish (100 eggs/dish), 0.5 ppm methylene blue was added to egg water (0.75 g calcium sulphate, 0.3 g instant sea salt, 1 L ultra-pure water), then, 2 h later, they were transferred into fresh egg water and incubated at 28.5 °C. 

### 2.3. Particulate Matter (PM) Preparation and Treatment

Particulate matter (PM) stock solution (5 mg/mL) was prepared with egg water. Then, 6–7 h old embryos were transferred to a 24 multi-well plate with 15 embryos per well and treated with the indicated concentrations of PM. 

### 2.4. Mortality, Heartbeat, Hatching, and Malformation Rate Analysis

PM (200 and 400 µg/mL)-treated embryos were observed at 24, 48, 72, and 96 h after the treatment using a fluorescence microscope (SZX16, Olympus Corporation, Tokyo, Japan) to determine the mortality, hatching, and malformation rate in zebrafish embryos. The heartbeat was observed at 72 hpf (hours post fertilization). Heartbeat was observed and measured for 3 min under a stereomicroscope, and the results were calculated as an average value for 1 min.

### 2.5. Body Length, Yolk Sac Area, and Cardiac Area Analysis

At 72 h after PM (200, 400 µg/mL) treatment, the embryos were observed by using a microscope (SZX16, Olympus Corporation, Tokyo, Japan). Total body length was measured by drawing a straight line from head to tail, while total yolk sac and cardiac area were measured by demarcating the respective areas using Olympus cellSens Standard software (Version 3.1, Olympus Corporation, Tokyo, Japan).

### 2.6. Eye Morphology Analysis

Eye morphology was observed according to a previously described method with slight modifications [24]. Briefly, at 72 h after PM (200, 400 µg/mL) treatment, the eyes of the zebrafish embryos were observed by using a microscope (SZX16, Olympus Corporation). Vertical eye diameter and horizontal eye diameter were measured by drawing straight lines from the y and x axes, respectively. Total eye area and total eye lens area were measured by demarcating the respective areas using Olympus cellSens Standard software (Version 3.1).

### 2.7. Total Reactive Oxygen Species (ROS) Evaluation

In order to determine the whole-body and eye ROS content in zebrafish embryos, 2′,7′-dichlorodihydrofluorescein diacetate (H_2_DCF-DA, 10 µM) was added to the 72 hpf zebrafish embryos from each treatment group and stained at 28 °C for 1 h in dark conditions. After that, the embryos were washed with egg H_2_O 3 times and observed by using a fluorescence microscope (SZX16, Olympus Corporation). The fluorescence intensities of captured images were analyzed using ImageJ software v1.8.0 (NIH, MD, WI, USA).

### 2.8. RT-qPCR

Real-time quantitative PCR analysis was conducted according to a previously described method with slight modifications [25]. Total mRNA from five embryos was extracted using Trizol reagent (TaKaRa Bio Inc., Kusatsu, Japan). Briefly, the zebrafish embryos were washed using cold PBS, and 1 mL of Trizol reagent was added, and the embryos were homogenized for 5 min on ice. Then, 200 µL of chloroform was added and kept for 5 min at room temperature (RT). Then, it was centrifuged at 12,000× *g* for 15 min at 4 °C. After 15 min, the clear supernatant (around 400 µL) was transferred to a new 1.5 mL e-tube, and 500 µL of iso-propanol was added and mixed gently a few times and kept for 10 min at RT. After that, it was centrifuged at 12,000× *g* for 10 min at 4 °C to precipitate the RNA. Then, the supernatant was removed, and the RNA pallet was cleaned using 75% cold ethanol at 7500× *g* for 5 min at 4 °C. The RNA concentration was measured using a Nano Drop (Thermo Scientific, Waltham, MA, USA). cNDA was prepared using TaKaRa 1st strand cDNA synthesis kit (TaKaRa Bio Inc.) according to the manufacturer’s instructions. Real-time quantitative PCR was performed by using SYBR Green in a Bio Rad real-time PCR system (Bio-Rad laboratories Inc, Hercules, CA, USA). Relative expression levels of antioxidant-related genes, including catalase (*cat*) and superoxide dismutase 2 (*sod 2*); photoreceptor cell development- and pigmentation-related genes, including rhodopsin (*rho*), atonal bHLH transcription factor 8 (*atoh 8*), and visual system homeobox (*vsx 1*); eye-lens development-related genes, including crystallin alpha A (cryaa, crystallin, beta B1—*crybb1*); and eye embryogenesis-related genes, including paired box 6a (*pax6a*) and paired box 6b (*pax6b*), were measured. The internal controls, including β-actin 1 (*actb1*), glyceraldehyde-3-phosphate dehydrogenase (*gapdh*), tubulin alpha 1b (*tuba1b*), 18s ribosomal RNA (*18s rRNA*), beta-2-microglobulin (*b2m*), and elongation factor 1 alpha (*elfa*), confirmed the stability caused by PM exposure [26,27]. As shown in Appendix A, none of the housekeeping genes was affected by PM treatment. Beta-actin 1 (*actb1*) was used as the housekeeping gene in this study. Primer sequences are shown in Table 1 and Appendix A. The fold change for each gene was calculated by using the 2^−ΔΔCt^ method based on threshold cycle (CT) values.

### 2.9. Statistical Analysis

All the experimental data are expressed as the mean ± SD. One-way and two-way analyses of variance (ANOVA) were conducted to determine the significant differences among experimental groups with Tukey’s multiple mean comparison test. *p* < 0.05 was considered statistically significant. All statistical analyses were performed by using GraphPad Prism software version 8 (GraphPad Software, Boston, MA, USA).

## 3. Results

### 3.1. Particulate Matter (PM) Induced Toxicity in Zebrafish Embryos

Particulate matter (PM) is well known as a toxic environmental substance. We, therefore, first conducted an experiment to determine whether PM was toxic to zebrafish embryos. PM treatment significantly increased the mortality (Figure 1a) and heartbeat (Figure 1b), and decreased the hatching rate (Figure 1c,d) in a dose-dependent manner, but eventually, all groups of embryos were hatched at 96 hpf (hours post fertilization, Figure 1d). In addition, PM induced developmental malformations in a dose- and time-dependent manner (Figure 1e,f). At the highest concentration of PM (400 µg/mL), the mortality was significantly increased up to 30%, but the developmental malformations significantly increased up to 100% compared to those of the control group at 96 hpf. The developmental malformations caused by PM toxicity were characterized by severe forms of pericardial edema (blue arrow), yolk sac edema (red arrow head), and trunk curvature (yellow arrow) (Figure 1f). Because of the severe dose-dependent edema occurrence in PM, the body length, pericardial, and yolk sac size were determined. As expected, PM inducement slows the growth of body length while increasing the yolk sac and pericardial area (Figure 1g–j), suggesting that PM induced toxicity in zebrafish embryos.

### 3.2. Particulate Matter (PM) Induced Eye Development Alteration in Zebrafish Embryos

Direct eye exposure to PM affects not only fish but also humans; therefore, the PM toxicity analysis focused on eye development in zebrafish embryos. The total eye size, total lens size, and vertical and horizontal eye diameters are presented in Figure 2a. The total eye area (Figure 2b,c), eye lens area (Figure 2b,d), and horizontal and vertical eye diameters (Figure 2b,e,f) were significantly decreased in PM-treated embryos compared to those of the control in a dose-dependent manner, demonstrating that the PM induced eye size reduction in zebrafish embryos.

### 3.3. Particulate Matter (PM) Increased Oxidative Stress in Zebrafish Embryos

Oxidative stress is one of the representative toxicity markers; thus, first, a H_2_DCF-DA assay was conducted at 72 hpf after zebrafish embryos were exposed to 200 and 400 µg/mL PM concentrations to determine the ROS (reactive oxygen species). The whole-body (Figure 3a,b) and eye (Figure 3c,d) ROS overexpression was significantly increased in zebrafish embryos in a dose-dependent manner, which was characterized by the increased relative fluorescence intensities in the whole-body and eye regions of PM-treated zebrafish embryos. Next, the mRNA expression of antioxidant enzymes was determined. As the results show, *cat* and *sod2* mRNA expression levels were downregulated in a PM-concentration-dependent manner (Figure 4), suggesting that PM caused oxidative stress.

### 3.4. Particulate Matter (PM) Induced mRNA Expression Alteration in Eyes of Zebrafish Embryos

Next, we investigated whether the reduction in eye size in response to PM toxicity was related to photoreceptor cell development and ocular pigmentation. The photoreceptor-cell-development- and eye-pigmentation-related mRNA expression of atonal bHLH transcription factor 8 (*atoh8*), rhodopsin (*rho*), and visual system homeobox (*vsx1*) were downregulated with PM in a dose-dependent manner (Figure 5a–c). These results are the same for eye size reduction. The lens, together with the cornea, is a special transparent tissue that focuses light onto the retina [28]. As the size of the eye increases, the focal length of the lens becomes longer, and more light is received even in dark places [29,30]. Therefore, PM-induced reduction in eye size is probably accompanied by a decrease in photoreceptor cell development, as shown in our results, reducing light intensity control in the lens, and thus reducing visual acuity. On the other hand, the mRNA expression of the eye-lens-development-related gene crystall in alpha A (*cryaa*) and paired box 6a (*pax6a*) and paired box 6b (*pax6b*), which are eye-embryogenesis-related genes, were downregulated in PM-exposed embryos compared to the control group, but the eye-lens-development-related gene crystallin beta B1 (*crybb1*) was up-regulated in PM-treated embryos (Figure 5d–g). These results suggest that PM exposure can affect zebrafish eye development by downregulating and overexpressing various eye-specific genes.

## 4. Discussion

Particulate matter (PM) is a concerning air pollution factor which leads to mortality and morbidity by causing severe diseases in humans such as cardiovascular [31,32], digestive [33], and respiratory diseases [34,35]. Due to the small particle size of PM, it can easily penetrate the human body through the respiratory and digestive tract, which leads to severe diseases, causing mortality and morbidity in humans [36]. Most previous studies have mainly focused on the adverse effects of PM on cardiovascular and respiratory systems, but the visual system is one of the systems in which the organs are directly exposed to air pollution. The most sensitive organ, the eye, is regularly exposed to PM, which can cause vision defects. Even though eye diseases are not life-threatening, they can drastically reduce quality of life [37]. Therefore, in our study, we used the zebrafish embryo, an aquatic animal model that satisfies the 3Rs, that is, reduction, replacement, and refinement [38,39], to evaluate PM-induced embryonic developmental toxicity by focusing on the visual system. We were able to demonstrate that PM has the potential to induce embryonic and eye developmental toxicity by altering the hatching rate, inducing severe malformations, stunting growth, and reducing eye size.

The zebrafish is becoming a popular model organism in toxicity experiments due to its advantages over other mammalian models such as mice. Mammalian models are prone to high costs, time wasting, and ethical issues, while the zebrafish is low cost, has fewer ethical issues, and has genetic compatibility with humans. It has recently emerged as an alternative animal model to study various human diseases caused by environmental hazards like PM [21]. PM was able to induce embryonic toxicity by increasing mortality, decreasing the hatching rate, and increasing the heartbeat in a dose-dependent manner. In the highest tested concentration of PM (400 µg/mL), the hatching rate was reduced by less than 10% to around 48 hpf compared to that of the control group, while mortality increased by nearly 30% to around 96 hpf. PM caused cardiac developmental toxicity in zebrafish embryos via the AhR and wnt/β-catenin pathways [40]. As part of the embryonic developmental toxicity, our results indicated that PM has the capacity to induce severe malformations during the early developmental stages of zebrafish embryos. At around 72 hpf, PM-exposed zebrafish embryos showed severe malformations, such as cardiac edema, yolk sac edema, trunk curvature, and stunted growth through a decrease in the total body length, in a dose-dependent manner. Indeed, the severe malformations in the cardiac region and yolk sac region demonstrate the increase in the total cardiac area and total yolk sac area, respectively. 

As the main aspect of this study, we focused on PM-induced eye developmental toxicity in zebrafish embryos. We found that the general eye size of zebrafish embryos tended to decrease in a dose-dependent manner, indicating the early development of microphthalmia in zebrafish eyes. This can also induce reduce the total eye area, total eye lens area, and horizontal and vertical eye diameters. Various factors, like ionization radiation exposure and toxic chemical (e.g., ethanol, heavy metals) exposure, can also induce microphthalmia during the early development of zebrafish embryos [36]. Some PM contains PAHs, which are derived from the incomplete burning of fossil fuels and organic materials and tend to significantly reduce the eye size of early developing zebrafish embryos by reducing their light sensitivity by up to 40% [24]. Therefore, our results suggest that various hazardous substances, such as PAHs, tend to induce eye developmental toxicity in zebrafish embryos.

Even though the mechanism of PM toxicity has not been clearly illustrated, ROS production plays a critical role in developmental toxicity in zebrafish embryos. ROS overproduction leads to oxidative stress, due to molecular, cellular, and tissue damage, which causes non-cancerous and cancerous ophthalmic diseases such as glaucoma, cataracts, inflammatory uveitis, lymphoma, retinoblastoma, and melanoma [41]. Therefore, we investigated the whole-body and eye-region ROS expression. Our results indicate that the ROS expression in these regions was significantly induced through PM exposure in a dose-dependent manner. The eyes are directly exposed to environmental hazardous materials, which cause ROS, and this oxidative stress caused by excessive ROS can be devastating during the early development stages of the eye, when cells proliferate and differentiate to create a developed eye structure. Moreover, the overproduction of ROS in embryonic eyes can cause cell death and eye size reduction [36]. Therefore, our results suggest that PM-induced excessive ROS production causes abnormalities in zebrafish embryonic and eye development. The underlying mechanisms of PM toxicity were confirmed via antioxidant enzyme mRNA expression levels in zebrafish embryos. The mRNA expression levels of *cat* and *sod2* were clearly downregulated in PM-treated embryos in a dose-dependent manner; the downregulation of these antioxidative genes causes an accumulation of excessive ROS in cells and tissues, which then leads to cell death [42].

In this study, as we specifically focused on the effects of PM on zebrafish eyes, the mRNA levels of eye-specific genes were also examined. Photoreceptor-cell-development- and pigmentation-related genes (*atoh8*, *vsx1*, and *rho*), eye-lens-development-related genes (*cryaa* and *crybb1*), and eye-embryogenesis-related genes (*pax6a* and *pax6b*) were examined after exposure to PM. Apart from *crybb1*, all other genes were clearly downregulated by PM in a dose-dependent manner. Gene knockdown studies have already identified that the endogenous functions of *pax6a*, *pax6b*, *vsx1*, *otx2*, and *rx1* are clearly involved in the eye morphogenesis of vertebrates, and changes in the expression levels of these genes could cause various eye diseases, like microphthalmia [43,44]. Therefore, we hypothesized that the general eye size reduction caused by PM exposure, which indicates the early onset of microphthalmia in zebrafish embryos, could be the result of the downregulation of eye-embryogenesis-related genes, such as *pax6a* and *pax6b*. Daytime vision in humans is mediated primarily by cone photoreceptors in the retina, and vision in larval zebrafish is also mediated almost exclusively by cone photoreceptors [45]. In addition, the degeneration of photoreceptors is the most common form of blindness in the world and involves the loss of vision due to the dystrophy and/or death of retinal photoreceptors [22]. Various genes, such as *atoh8* and *vsx1*, play a critical role in photoreceptor cell development in zebrafish embryos. The downregulation of these genes can cause abnormalities in the pigment cell layer, which lead to complications in light sensitivity [46]. Our results also demonstrate the downregulation of *atoh8* and *vsx1* mRNA expression levels in PM-exposed zebrafish embryos, indicating that PM can affect photoreceptor cell development in zebrafish embryos, which leads to reduced light sensitivity in zebrafish embryos. This is further confirmed via the downregulation of *rho* (rhodopsin), which is a light-sensitive opsin protein in the pigment cell layer of zebrafish embryos. Changes in opsin proteins could lead to eye diseases, such as blindness [47]. Cataracts in the eye are characterized by the opacity of the eye lens in vertebrates. Mutations in crystallin-encoding genes such as *cryaa* and *crybb1* cause cataracts in mammals, including humans [48]. Our results indicated the downregulation of *cryaa* mRNA expression and too much upregulation of *crybb1* mRNA expression in PM-exposed embryos, which suggest that abnormalities in the zebrafish eye lens may increase the incidence of cataracts during the early development of zebrafish embryos.

To summarize, in the early stages of zebrafish development, PM exhibited various toxicities, such as a decreased hatchability, increased mortality, malformations, and increased heart rate. It was demonstrated that PM affects eye development by increasing ROS production and regulating the expression of antioxidant genes, photoreceptor-cell-development and pigmentation genes, eye-embryogenesis genes, and eye-lens-development genes. Despite this, our results indicate that PM downregulates and upregulates the expression levels of various eye-specific genes, and further deep analysis would be needed to clearly understand the underlying mechanisms of various eye diseases.

## 5. Conclusions

In summary, PM showed time- and dose-dependent embryo toxicity and morphological toxicity, such as a reduced hatchability, increased mortality, malformations, and heart rates in zebrafish embryos. In particular, PM increased ROS production in the whole body and eyes of zebrafish embryos and reduced the expression of antioxidant genes. Our results suggest that PM treatment affected eye development by regulating the expression of photoreceptor-cell-development and pigmentation genes, eye-embryo-generation genes, and eye-lens-development genes. All this suggests that further research is required to analyze mechanisms related to various eye diseases, but our study shows that zebrafish embryos can be used as an alternative model for evaluating PM-induced eye development toxicity *in vivo*.

## Figures and Tables

**Figure 1 toxics-12-00059-f001:**
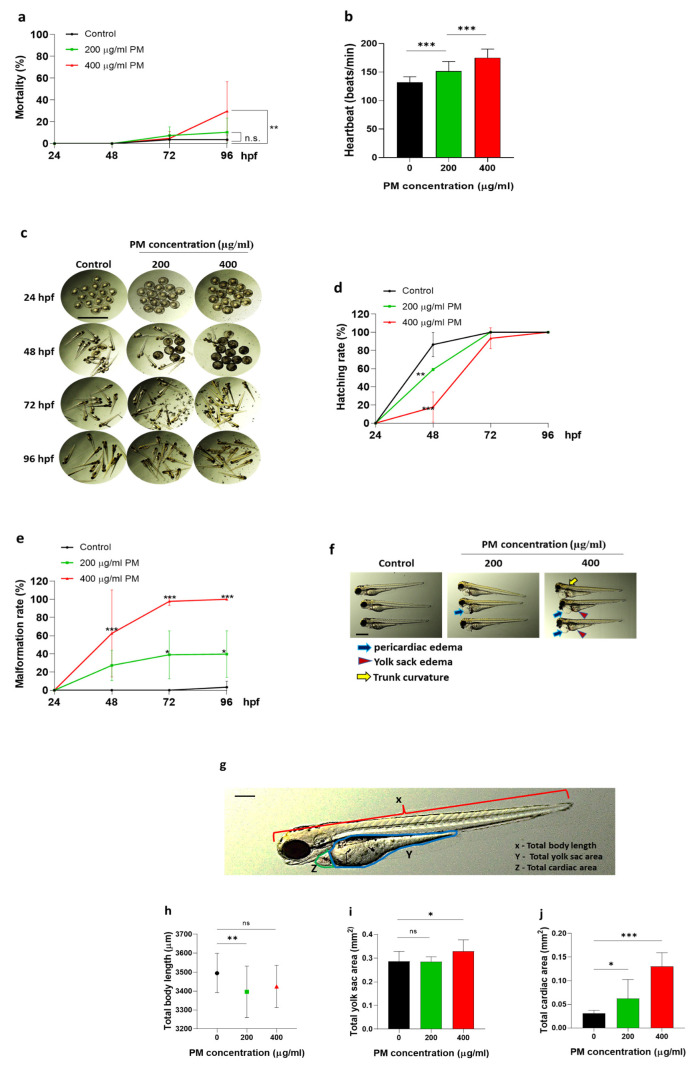
Particulate matter (PM) induced toxicity in zebrafish embryos. Zebrafish embryos aged 6~7 h post fertilization (hpf) were incubated with 200(green) and 400(red) µg/mL of PM up to 96 hpf. PM increased the (**a**) mortality and (**b**) heartbeat in a dose-dependent manner. (**c**,**d**) A decreased hatching rate was observed in PM in a dose-dependent manner. (**e**,**f**) The malformation rate increased with PM in a dose-dependent manner. (**g**–**j**) The total body length, yolk sac area, and cardiac area were observed as malformation parameters. Data represent the mean ± S.D. SD: standard deviation, scale bar: 1 mm, ns: not significant * *p* < 0.05, ** *p* < 0.01, *** *p* < 0.001.

**Figure 2 toxics-12-00059-f002:**
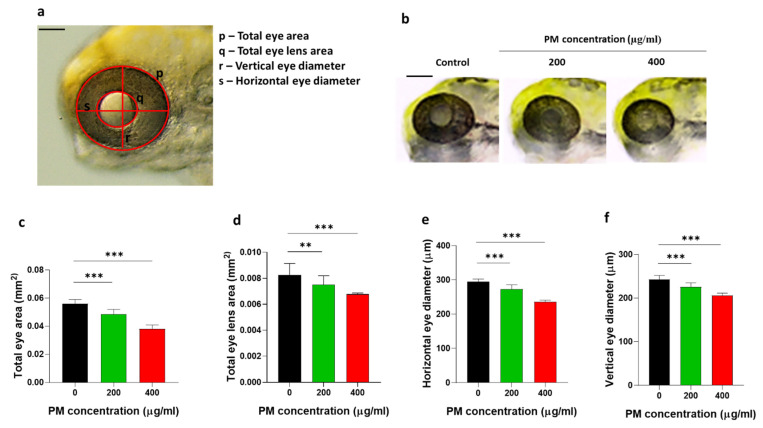
Particulate matter (PM) induced eye toxicity in zebrafish embryos. Zebrafish embryos aged 6~7 h post fertilization (hpf) were incubated with 200 and 400 µg/mL of PM. (**a**) The anatomical parameters of the eye were measured at 72 hpf. (**b**) Representative images of reduced overall eye size in the embryos that were incubated with the indicated concentrations of PM. (**c**) The total eye area, (**d**) total eye lens area, (**e**) horizontal eye diameter, and (**f**) vertical eye diameter decreased in the embryos that were incubated with the indicated concentrations of PM. Data represent the mean ± SD. SD: standard deviation, scale bar: 1 mm, ** *p* < 0.01, *** *p* < 0.001.

**Figure 3 toxics-12-00059-f003:**
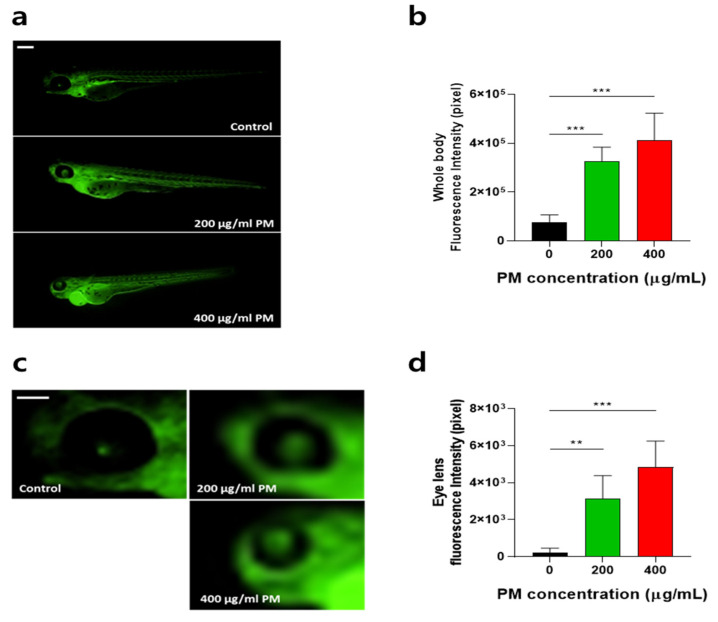
Particulate matter (PM) induced whole-body and eye reactive oxygen species (ROS) generation in zebrafish embryos. Zebrafish embryos aged 6~7 h post fertilization (hpf) were incubated with the indicated concentrations of PM. The whole-body and eye ROS were significantly increased. (**a**,**c**) Representative images and (**b**,**d**) quantitative analysis of ROS generation from the images. Data represent the mean ± SD. SD: standard deviation, ** *p* < 0.01, *** *p* < 0.001. Scale bar: 1 mm.

**Figure 4 toxics-12-00059-f004:**
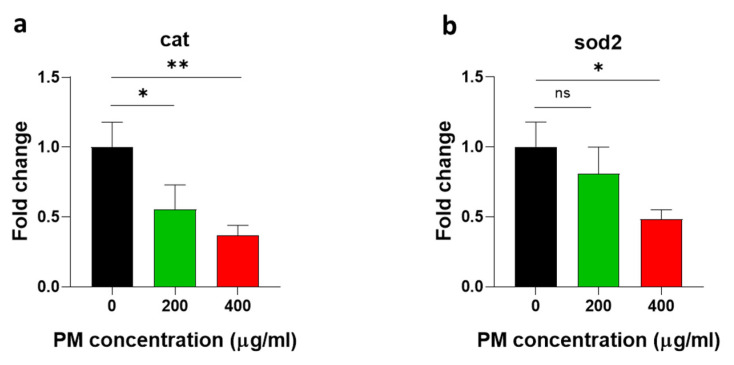
Particulate-matter (PM)-induced alterations in antioxidant enzyme mRNA expression. Zebrafish embryos aged 6~7 h post fertilization (hpf) were incubated with the indicated concentrations of PM. The mRNA expression of (**a**) catalase (*cat*) and (**b**) superoxide dismutase 2 (*sod2*) were reduced through treatment with PM. Data represent the mean ± SD. SD: standard deviation, ns: not significant, * *p* < 0.05, ** *p* < 0.01.

**Figure 5 toxics-12-00059-f005:**
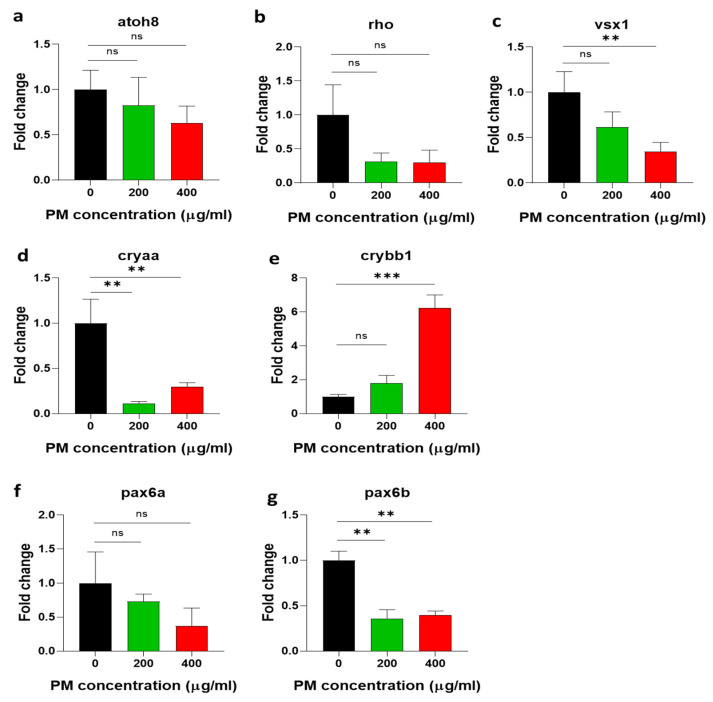
Particulate-matter (PM)-induced mRNA expression alterations in (**a**–**c**) photoreceptor cell development and pigmentation, (**d**,**e**) eye lens development, and (**f**,**g**) eye embryogenesis in zebrafish embryos. Zebrafish embryos aged 6~7 h post fertilization (hpf) were incubated with the indicated concentrations of PM. *atoh8*: atonal bHLH transcription factor 8, *rho*: rhodopsin, *vsx1*: visual system homeobox, cryaa: crystallin alpha A, *crybb1*: crystallin beta B1, *pax6a*: paired box 6a, *pax6b*: paired box 6b. Data represent the mean ± SD. SD: standard deviation, ns: not significant, ** *p* < 0.01, *** *p* < 0.001.

**Table 1 toxics-12-00059-t001:** Primers for real-time qPCR.

Gene		Nucleotide Sequence	Gene ID
*actb1*	Forward	5′ CTATGAGCTGCCTGACGGTC 3′	NM_131031.2
	Reverse	5′ ATGTCCACGTCGCACTTCAT 3′	
*atoh8*	Forward	5′ TACGGCCGTAGACATGAGGA 3′	NM_001079991.2
	Reverse	5′ CAACACAACCCGCTCCAAAG 3′	
*cryaa*	Forward	5′ CTTCCGCAACATCCTGGACT 3′	NM_152950.2
	Reverse	5′ TTTCTCCATGCTTGCCCTGG 3′	
*crybb1*	Forward	5′ CGATGCCAAGGAGAAGGGAG 3′	NM_173231.2
	Reverse	5′ CGCTCACAAACGTTCATGCA 3′	
*vsx1*	Forward	5′ TTTTCTCCCGAGCCACATCC 3′	NM_131333.1
	Reverse	5′ GGTGAAAACTGTCCTGTGCC 3′	
*pax6a*	Forward	5′ ACCTTCCTATGCAACCCAGC 3′	NM_131304.1
	Reverse	5′ GGCACTTGAACGGGTACAGA 3′	
*pax6b*	Forward	5′ GAACCAGAGACGACAAGCCA 3′	NM_131641.1
	Reverse	5′ TTGGCCATAGTGAAGCTGGG 3′	
*rho*	Forward	5′ GCCTTCCTCATCTGCTGGTT 3′	NM_131084.1
	Reverse	5′ AGGGTGGTGATCATGCAGTG 3′	
*sod2*	Forward	5′ CGTGTGCTAACCAAGACCCT 3′	NM_199976.1
	Reverse	5′ GGAAACGCTCGCTGACATTC 3′	
*cat*	Forward	5′ GTGCATGCATGACAACCAGG 3′	NM_130912.2
	Reverse	5′ CGCTCTCTCTCGGCTTCATT 3′	

*actb1*: actin beta 1, *atoh8*: atonal bHLH transcription factor 8, *cryaa*: crystallin alpha A, *crybb1*: crystallin beta B1, *vsx1*: visual system homeobox, *pax6a*: paired box 6a, *pax6b*: paired box 6b, *rho*: rhodopsin, *sod2*: superoxide dismutase 2, *cat*: catalase.

## Data Availability

All data underlying the results are available as part of the article.

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
