# Peer review of "Particulate Matter Induced Adverse Effects on Eye Development in Zebrafish (Danio rerio) Embryos"

_toxics, 2024, doi:10.3390/toxics12010059_

Round 1

Reviewer 1 Report

Comments and Suggestions for Authors

In this manuscript, the authors describe their study which sought to determine the impact of particulate matter (PM) exposure on proper overall development. The authors also focused on the impact of PM on eye development. Various end points were assessed including survival, hatching, growth, eye size, lens size, ROS markers, and expression of retina-specific genes.  The results indicate that developmental exposure to PM does impact overall as well as vision-specific development in zebrafish.  The project as a whole as high potential impact; however, it needs to be heavily edited for clarity before it is suitable for publication.

 Major:

1. The abstract is missing.

 2. What are environmental concentrations of PM?  What levels are known to cause problems in humans?  Building on this, how were the concentrations used chosen?  If the values used are not environmentally relevant, a rationale should be provided.

 3. English language needs to be corrected throughout the document.

 4. The discussion is basically a recap of the results, including figure citations. It should be rewritten to summarize the results rather than restate them and the figure citations should be removed.

 5. Lines 250-252. The authors repeatedly state ‘visionary system’.  I believe they mean ‘visual system’.

 6. Figure 2 caption (a) states ‘physiological parameters of the eye were measured…’  That is not accurate. Anatomical parameters were measured.

 7. Line 53 states “The zebrafish… is a widely used novel model organism…”.  I’m not sure it can be both. If it is widely used, then it would not be novel, correct?  While either term could be used, the text that follows indicates zebrafish are used in a diversity of studies, so ‘widely used’ is probably the best adjective here.

 8. Lines 292-294 state “Therefore, our results suggest that PM contains various hazardous material like PAHs tend to induce eye development toxicity…..”   Since the authors do not assess the content of the PM they used, they cannot conclude that they contain PAHs (or any other chemical).  Also, isn’t it possible that the irritant nature of the PM is causing the observed effects and not necessarily any chemicals that are present?

 9. Did the particulate matter remain in solution when it was administered to the fish? 

 10.Do the authors think the PM is having an effect by direct contact (i.e., on the outside of the eye) or through ingestion/cellular metabolism?  If ingestion, have the authors done any work showing that uptake of PM has occurred in their experiments?

 11. Line 341 – “opacification”  should be “opacity”

 12. The figures are very clear and easy to read.

Comments on the Quality of English Language

The English language needs to be significantly improved.  Word choice, grammar, sentence structure, and tense are all poor.

For example, "Eye directly exposure PM not only fish also human...." is used in the results.

Author Response

Manuscript Number: toxics-2775981-12-1
Particulate matter induced adverse effects on eye development in zebrafish (Danio rerio) embryos

Dear Reviewer

We appreciate the time you have taken to review this paper and for the opportunity to submit a revised manuscript. We have carefully considered all of your comments and have revised the article based on them, and we hope that the manuscript, after further careful revision, will meet your high standards. All revisions to the manuscript have been made and are summarized below.

  1. The abstract is missing.

[Response] Could you please carefully reread the article? The abstract was in the manuscript on the first page.

  1. What are environmental concentrations of PM? What levels are known to cause problems in humans? Building on this, how were the concentrations used chosen? If the values used are not environmentally relevant, a rationale should be provided.

[Response] Although it has been reported that PM has a serious impact on human health, the composition of PM varies, and it has not yet been fully established at what concentration it is harmful to humans or how harmful it is. We have therefore addressed this matter in the Introduction, lines 25-53.

The PM concentration we used was selected from the LC50 concentration that was most frequently identified among the concentrations listed in the literature. Then, the concentration at which LC50 can be obtained was set and used in the experiment.

  1. English language needs to be corrected throughout the document.

[Response] According to the reviewer`s suggestion, the revised manuscript’s English has been improved by an English editor of Toxics.

  1. The discussion is basically a recap of the results, including figure citations. It should be rewritten to summarize the results rather than restate them and the figure citations should be removed.

[Response] According to the reviewer`s comments, the figure citations were removed from the Discussion section.

  1. Lines 250-252. The authors repeatedly state ‘visionary system’. I believe they mean ‘visual system’.

[Response] We appreciate the reviewer’s detailed reading, and “visionary system” has been changed to “visual system” accordingly throughout the article.

  1. Figure 2 caption (a) states ‘physiological parameters of the eye were measured…’ That is not accurate. Anatomical parameters were measured.

[Response] We appreciate the reviewer’s recommendation; “physiological parameters of the eye were measured” has been changed to “Anatomical parameters of the eye were measured” accordingly.

  1. Line 53 states “The zebrafish… is a widely used novel model organism…”. I’m not sure it can be both. If it is widely used, then it would not be novel, correct? While either term could be used, the text that follows indicates zebrafish are used in a diversity of studies, so ‘widely used’ is probably the best adjective here.

[Response] We appreciate the reviewer’s detailed reading; the use of “novel” was deleted from the manuscript.

Line 55.

  1. Lines 292-294 state “Therefore, our results suggest that PM contains various hazardous material like PAHs tend to induce eye development toxicity…..” Since the authors do not assess the content of the PM they used, they cannot conclude that they contain PAHs (or any other chemical). Also, isn’t it possible that the irritant nature of the PM is causing the observed effects and not necessarily any chemicals that are present?

[Response] We appreciate the reviewer’s critical reading and agree with it; the description “Therefore, our results suggest that various hazardous substances such as PAHs tends to induce eye developmental toxicity in zebrafish embryos.” has been changed from the Discussion section.

Line 287-289.

  1. Did the particulate matter remain in solution when it was administered to the fish? 

[Response] The particulate matter was not completely dissolved. So, we mixed the particulate matter solution thoroughly right before use and used it in the experiment.

  1. Do the authors think the PM is having an effect by direct contact (i.e., on the outside of the eye) or through ingestion/cellular metabolism? If ingestion, have the authors done any work showing that uptake of PM has occurred in their experiments?

[Response] We think that particulate matter has complex adverse effects through both direct contact and ingestion. The reasons for this are as follows: (1) Particulate matter caused toxicity even though it was not completely dissolved. In other words, it can be assumed that the toxicity caused by exposure to particulate matter is caused by external contact. (2) The particulate matter that we used is a mixture that is not classified by size and composition. Therefore, there may be a mixture of particles of different sizes. It has been reported that smaller particle sizes allow penetration into cells (https://www.ncbi.nlm.nih.gov/pmc/articles/PMC6027495/). (3) Lastly, the yolk sac of zebrafish embryos acts as a nutrient, and the embryos absorb the yolk sac by themselves for survival. Although it has not yet been known why and how they absorb the yolk sac into their body, it has been reported that substances with high molecular weight are also absorbed through the process of yolk sac absorption. Therefore, we supposed that the particulate matter toxicity is caused by both direct contact and ingestion. A particulate matter ingestion experiment was not conducted.

  1. Line 341 – “opacification” should be “opacity”

[Response] We appreciate the reviewer’s detailed reading, and “opacification” has been changed to “opacity” accordingly.

Line 332.

Reviewer 2 Report

Comments and Suggestions for Authors

Toxics

Manuscript #2775981

Particulate matter induced adverse effects on eye development 2 in zebrafish (Danio rerio) embryos

The authors evaluated the toxic effects of particulate matter on the embryonic development of zebrafish, focusing on possible changes in the development of the larval eye. To do this, they exposed 6 to 7 hpf embryos to the pollutant in 2 concentrations dissolved in the medium. Analyzes carried out every 24 hours up to 96 hours after exposure show low lethality and sub-lethal changes such as yolk sac and pericardial edema with tachycardia and eye malformation. They found an association between ocular malformations and oxidative stress and modulation of genes responsible for ocular development.

Specific Comment:

1. The authors clarify that the objective of the work was:

.... evaluating the effects of PM on eye disease development and progression via the zebrafish embryo model. However, what was carried out in this study the evaluation of the toxic effects of PM on embryonic development, focusing especially on ocular development.

2. Could the authors clarify why they used 6 to 7 hpf embryos at a rate of 15 per group? Why didn't they use the OECD #236: Fish Embryo Acute Toxicity (FET) Test for toxicity analysis? This test is classically used to identify the toxicity of toxins, particulate matter, and nano particles.

3. Could authors describe the vivarium (University) of origin of the fish used in the study?

4. Describe how heartbeats were counted in 72 hpf larvae and indicate the method in the Reference List. Check: MacRae, C. A., & Fishman, M. C. (2002). Zebrafish: the complete cardiovascular compendium. Cold Spring Harbor symposia on quantitative biology, 67, 301–307. https://doi.org/10.1101/sqb.2002.67.301

5. In Figure 1, change the order of the figures: the heartbeat graph (b) can be placed together with the cardiac area measurement graph, which will be (j).

6. In the Discussion section, line 248, correct the sentence: leads to severe diseases by causing mortality and morbidity in humans

7. In the Discussion section, line 250, 251, 256 correct the word: visionary system by visual system

8. In the Discussion section, line 255 and 263, correct the sentence: zebrafish is an alternative aquatic animal model as it obeys the concept of 3Rs translated into reduction, replacement and refinement (Strahle et al., 2012; Vincent et al., 2015).

U. Strähle, S. Scholz, R. Geisler, P. Greiner, H. Hollert, S. Rastegar, A. Schumacher, I. Selderslaghs, C. Weiss, H. Witters, T. Braunbeck, Zebrafish embryos as an alternative to animal experiments—A commentary on the definition of the onset of protected life stages in animal welfare regulations, Reprod. Toxicol. 33 (2012) 128–132. https://doi.org/https://doi.org/10.1016/j.reprotox.2011.06.121.

F. Vincent, P. Loria, M. Pregel, R. Stanton, L. Kitching, K. Nocka, R. Doyonnas, C. Steppan, A. Gilbert, T. Schroeter, M.-C. Peakman, Developing predictive assays: the phenotypic screening “rule of 3”., Sci. Transl. Med. 7 (2015) 293ps15. https://doi.org/10.1126/scitranslmed.aab1201.

9. In the Discussion section, line 267-268, correct the sentence: with compared to

10. In the Discussion section, line 288, correct the sentence: can also be induce

11. In the Discussion section, line 293, correct the sentence: tend to induce. If there is a statistical difference with p< 0.05, the trend becomes a fact

12. In the Discussion section, line 294, the statement (visual impairment) will only be true if a change in locomotor activity is proven

13. In the Discussion section, line 300-303: improve the meaning of the sentence

14. In the Discussion section, line 304 fix: cause by

15. In the Discussion section, line 310 correct: PM toxicity was confirmed by antioxidant enzymes mRNA expression

16. In the Discussion section, line 295-314, correct the sentence to improve understanding

17. In the Discussion section, line 320, delete: caused

18. In the Discussion section, line 341-343, correct the sentence: the results show the reduction of the eye and genes responsible for the development of the retina and lens. However, it cannot be said that the larvae have cataracts.

19. In the References section, references 39 and 40 are the same: delete the duplicate

20. Authors could cite more current references, from 2019 to 2023

Comments on the Quality of English Language

the text need to be revised to improve understanding of the message the authors want to convey

Author Response

Manuscript Number: toxics-2775981-12-1
Particulate matter induced adverse effects on eye development in zebrafish (Danio rerio) embryos

Dear Reviewer

We appreciate you taking the time to review this paper and for the opportunity to submit a revised manuscript. We have carefully considered your comments and have revised the manuscript accordingly. We hope that, after further careful revision, it now meets your high standards. All revisions to the manuscript have been made and are summarized below.

  1. The authors clarify that the objective of the work was: .... evaluating the effects of PM on eye disease development and progression via the zebrafish embryo model. However, what was carried out in this study the evaluation of the toxic effects of PM on embryonic development, focusing especially on ocular development.

[Response] We appreciate the reviewer’s suggestion. The aim of this study is to establish a model system for eye diseases caused by environmental pollutants, particularly PM. The Introduction has been revised to better convey this objective.

Lines 62-66.

  1. Could the authors clarify why they used 6 to 7 hpf embryos at a rate of 15 per group? Why didn't they use the OECD #236: Fish Embryo Acute Toxicity (FET) Test for toxicity analysis? This test is classically used to identify the toxicity of toxins, particulate matter, and nano particles.

[Response] The OECD test method does not mention fish rearing area or water capacity. In our experiment, we established conditions like the developmental rate of fish in the control group under natural conditions; hence, this is why we used 15 embryos per group. Additionally, because three independent experiments were conducted in this study, the total number of embryos used was 45.

More importantly for the current study, the main purpose of the model system that we sought to establish through this study is to explore functional materials or medicines that can prevent eye diseases caused by PM. Therefore, we wanted to find a PM concentration that could cause disease rather than a PM concentration that has a high mortality.

  1. Could authors describe the vivarium (University) of origin of the fish used in the study?

[Response] The fish origin is described in the Materials and Methods section.

Line 75.

  1. Describe how heartbeats were counted in 72 hpf larvae and indicate the method in the Reference List. Check: MacRae, C. A., & Fishman, M. C. (2002). Zebrafish: the complete cardiovascular compendium. Cold Spring Harbor symposia on quantitative biology, 67, 301–307. https://doi.org/10.1101/sqb.2002.67.301.

[Response] According to the reviewer`s comments, we have detailed how we calculated heartbeats. However, we were unable to confirm our method for calculating heartbeats in the references suggested by the reviewer, so we did not include it in the References.

Lines 93-95.

  1. In Figure 1, change the order of the figures: the heartbeat graph (b) can be placed together with the cardiac area measurement graph, which will be (j).

[Response] We appreciate the reviewer’s detailed consideration; however, we did not modify Figure 1 for the following reason:

The results of Figure 1 are ordered as below.

  1. Basic toxicity assessment such as survival rate and heartrate.
  2. Organized in order of observation of external changes due to particulate matter toxicity.

Heartrate is an indicator that changes due to toxicity even if no malformation occurs. After careful consideration, we have decided to retain the current order.

  1. In the Discussion section, line 248, correct the sentence: leads to severe diseases by causing mortality and morbidity in humans.

[Response] The sentence has been changed according to the reviewer`s suggestion.

Line 241-242.

  1. In the Discussion section, line 250, 251, 256 correct the word: visionary system by visual system.

[Response] We appreciate the reviewer’s detailed reading, and “visionary system” has been changed to “visual system” accordingly throughout the article.

For example, lines 218, 244, 250 .

  1. In the Discussion section, line 255 and 263, correct the sentence: zebrafish is an alternative aquatic animal model as it obeys the concept of 3Rs translated into reduction, replacement and refinement (Strahle et al., 2012; Vincent et al., 2015).

[Response] We appreciate the reviewer’s detailed reading; the Discussion section has been greatly improved based on these suggestions, and the References section has been updated as well.

Lines 247-250.

  1. In the Discussion section, line 267-268, correct the sentence: with compared to 

[Response] We appreciate the reviewer’s detailed reading and have changed the sentence to "compared to that of the control".

Line 261.

  1. In the Discussion section, line 288, correct the sentence: can also be induce

[Response] We appreciate the reviewer’s detailed reading and have changed it to "can also induce".

Line 283.

  1. In the Discussion section, line 293, correct the sentence: tend to induce. If there is a statistical difference with p< 0.05, the trend becomes a fact

[Response] We appreciate the reviewer’s detailed reading and have changed it to "tends to induce".

Line 285-286.

  1. In the Discussion section, line 294, the statement (visual impairment) will only be true if a change in locomotor activity is proven

[Response] The "visual impairment" was deleted.

Line 288.

  1. In the Discussion section, line 300-303: improve the meaning of the sentence

[Response] The sentence was revised based on the reviewer`s suggestion.

Lines 295-298.

  1. In the Discussion section, line 304 fix: cause by

[Response] Thank you for your comment; it has been changed to “due to”.

Line 292.

  1. In the Discussion section, line 310 correct: PM toxicity was confirmed by antioxidant enzymes mRNA expression

[Response] Thank you for the detail reading, we changed it to "[The underlying mechanisms of] PM toxicity were confirmed by antioxidant enzyme mRNA expression".

Line 303.

  1. In the Discussion section, line 295-314, correct the sentence to improve understanding

[Response] We appreciate the reviewer’s detailed reading. These sentences have been modified to improve understanding.

Lines 297-307

  1. In the Discussion section, line 320, delete: caused

[Response] We appreciate the reviewer’s detailed reading; "caused" has been deleted.

Line 312.

  1. In the Discussion section, line 341-343, correct the sentence: the results show the reduction of the eye and genes responsible for the development of the retina and lens. However, it cannot be said that the larvae have cataracts.

[Response] We appreciate the reviewer’s critical reading and agree; we suggest that abnormalities in the zebrafish eye lens may increase the incidence of cataracts during the early development of the zebrafish embryo.

Lines 333-337.

  1. In the References section, references 39 and 40 are the same: delete the duplicate

[Response] We appreciate the reviewer’s detailed reading; duplicate references have been deleted and the References section has been updated accordingly.

  1. Authors could cite more current references, from 2019 to 2023

[Response] According to the reviewer`s suggestion, more current references have been added.

Round 2

Reviewer 1 Report

Comments and Suggestions for Authors

The authors have addressed all reviewer comments satisfactorily.  The paper is also much easier to read with the English language edits.

Reviewer 2 Report

Comments and Suggestions for Authors

The authors took care to review the text of the manuscript and a better result was found, allowing its publication.